# Tau; One Protein, So Many Diseases

**DOI:** 10.3390/biology12020244

**Published:** 2023-02-03

**Authors:** Parisa Tabeshmehr, Eftekhar Eftekharpour

**Affiliations:** Spinal Cord Research Centre, Department of Physiology and Pathophysiology, University of Manitoba, Winnipeg, MB R3E 0J9, Canada

**Keywords:** tau, tauopathy, neurodegeneration, Alzheimer’s disease, dementia, aging, MAPT, cytoskeleton

## Abstract

**Simple Summary:**

Tau is an important protein for maintaining the shape and normal function of nerve cells. There are many diseases that are identified by changes in this protein, yet examination of tau in these diseases shows robust differences in tau modifications. A question arises from these observations on whether tau can be used as a therapeutic target. In this review, we aimed to provide a general overview of tau-physiology and pathophysiology in neurodegenerative diseases and describe the current approaches for diagnosis and experimental/clinical trials. This review is intended to enhance the understanding of graduate students specializing in neurobiology.

**Abstract:**

Tau, a member of the microtubule-associated proteins, is a known component of the neuronal cytoskeleton; however, in the brain tissue, it is involved in other vital functions beyond maintaining the cellular architecture. The pathologic tau forms aggregates inside the neurons and ultimately forms the neurofibrillary tangles. Intracellular and extracellular accumulation of different tau isoforms, including dimers, oligomers, paired helical filaments and tangles, lead to a highly heterogenous group of diseases named “Tauopathies”. About twenty-six different types of tauopathy diseases have been identified that have different clinical phenotypes or pathophysiological characteristics. Although all these diseases are identified by tau aggregation, they are distinguishable based on the specific tau isoforms, the affected cell types and the brain regions. The neuropathological and phenotypical heterogeneity of these diseases impose significant challenges for discovering new diagnostic and therapeutic strategies. Here, we review the recent literature on tau protein and the pathophysiological mechanisms of tauopathies. This article mainly focuses on physiologic and pathologic tau and aims to summarize the upstream and downstream events and discuss the current diagnostic approaches and therapeutic strategies.

## 1. Tauopathies: Types and Importance

Tauopathies are a group of progressive neurodegenerative diseases characterized by tau protein inclusions in the human brain. Clinically, there are twenty-six different types of tauopathies that are recognized by complicated neurodegenerative symptoms, including dementia. A series of cognitive/behavioral and memory deficits manifest in individuals affected by tauopathies [1]. From a clinical pathology perspective, those neurodegenerative diseases in which tau protein plays the main pathophysiological role are classified as “primary tauopathies”. Pick’s disease (PiD), progressive supranuclear palsy (PSP), argyrophilic grain disease (AGD), primary age-related tauopathy (PART), neurofibrillary tangle dementia (NTD-dementia) and corticobasal degeneration (CBD) are several examples of primary tauopathies. On the other hand, for the “secondary tauopathies” tau protein aggregation is not the major neurodegenerative mechanism. For instance, Alzheimer’s disease (AD), the most common form of dementia, is known as a secondary tauopathy in which tau aggregation co-exists with accumulation of β-amyloid peptide (Aβ) [2]. Chronic traumatic encephalopathy (CTE) is another form of secondary tauopathy in which tau depositions are present in neurons, astrocytes and neurites around the blood vessels but are associated with other pathophysiological features [3].

According to the World Health Organization, nearly 50 million people are currently suffering from dementia, and each year, around 10 million new cases are diagnosed with dementia. It is predicted that by 2050, the number of patients will approximately reach 140 million worldwide. Due to the debilitating nature of dementia, these patients require constant care, which significantly impacts their careers, their families and society. In 2017, WHO proposed a “Global action plan on the public health response to dementia 2017–2025” in which dementia has been identified as a public health priority [4]. This rationalizes the need for better understanding all aspects of dementia’s pathophysiology, including tauopathies.

## 2. Tau

Tau protein, a member of the microtubule-associated protein family, is encoded by the microtubule-associated protein tau gene (MAPT) (Gene ID: 4137), located on human chromosome 17q21.31. Tau is primarily responsible for maintaining microtubules’ stability and promoting their assembly in axons. In the central nervous system (CNS), tau is mostly found in cortical and hippocampal neurons and, to a lesser degree, in astrocytes and oligodendrocytes [5]. Tau is not specific to the CNS, but it is also found in the peripheral nervous system (PNS), both intracellularly and extracellularly. The presence of tau protein has been reported in the interstitial fluid and cerebrospinal fluid (CSF) [6]. Structurally, there are four tau protein domains, including the N- and C-terminal domains, the proline-rich domain and the microtubule-binding domain. The proline-rich domain participates in the cell signaling process, as it is particularly the target of protein kinases. Phosphorylation of serin and threonine residues, which are mainly located in this domain can affect tau binding affinity in the microtubule-binding domain. The N-terminal domain may regulate the distance between microtubules. In contrast, the C-terminal domain is involved in microtubule polymerization [7].

The MAPT gene contains 16 exons [8], but its complex and highly regulated splicing results in a variety of developmental stage-specific messenger RNA (mRNA) species, which are mostly present in the brain tissue. In the adult human brain, this alternative splicing generates six isoforms, ranging from 352 to 441 amino acids [9]. Alternative splicing in exons 2 and 3 leads to three different isoforms, including 0N, 1N and 2N. Two additional isoforms, 3R and 4R, are also generated when splicing occurs in exon 10, resulting in the production of two proteins with three and four microtubule-binding domains, respectively. The impact of MAPT alternative splicing on human brain development and pathology has not been adequately investigated and may prove to be important. 

Expression levels of various tau isoforms reflect neuronal maturation state and axonal growth capability. Lower levels of tau expression have been reported in immature regions of the human brain, such as the ganglionic eminence and the rhombic lip, while tau expression pattern and splicing are slightly different in more mature brain regions [10]. Additionally, in the human prenatal brain, an abrupt shift happens in MAPT exons 2 and 10 expressions. This shift is evolutionary conserved and can be a crucial step in the transition to mature neurons. Exon 3 undergoes small temporal variations compared to the exons 2 and 10 [10]. Specifically, during early embryonic stages, the predominant tau isoform is 0N3R. Healthy adult human brains contain a balanced level of 4R and 3R isoforms. The other expressing human tau isoforms include 1N3R, 2N3R, 0N4R, 1N4R and 2N4R; the latter is the full-length form of tau protein. The 2N4R isoform contains both inserts of exons 2 and 3 in the N-terminal domain and all four microtubule-binding regions in the C-terminal domain. Collectively, this longest tau isoform includes all 441 amino acids [11], (Figure 1).

## 3. Physiologic Tau

Tau is an important component of the neuronal cytoskeletal compartment. The cellular skeleton is mainly composed of microtubules and is vital for the regulation of critical cell processes, including maintaining the cell shape, proper cell division, and healthy intracellular transport of organelles. In neurons, microtubular organization is critical for axonal stability and the trafficking of materials and organelles to and from the cell body. The dynamic of the cytoskeleton is highly dependent on microtubule-associated proteins (MAPs) [12], and amongst these, tau protein plays a critical role. Weingarten et al., in their milestone discovery, were able to identify tau protein as an essential factor for microtubule polymerization and microtubule assembly and dynamics [13]. As the microtubule dynamic network is essential for neurite formation and axonal pathfinding, the importance of tau protein has also been reported during neuronal development and CNS maintenance [14,15]. While tau protein expression is sharply increased during embryonic stages, this seems to reach a plateau in the mature rodent brain [16]. Authors showed that post-translational modification (PTM) of tau and its site-specific phosphorylation were differentially regulated between developing and mature brains [16]. While some residues were more phosphorylated in mature brain, overall, there was a decrease in tau phosphorylation in mature brain that might be due to an increase in tau-specific phosphatases [16].

The microtubule-binding inserts in the C-terminal domain of tau protein are the critical regions not only for binding to the microtubules but also for interacting with the actin filaments in the cytoskeleton structure [17,18]. Actin bundling is regulated by short fragments of the microtubule-binding domain in the C-terminal region of tau protein. In cell-free experiments, a synthetic version of the C-terminal region was able to bind to both monomeric and filamentous actin, but this interaction did not result in actin bundling. The authors showed that MAP2 and tau are capable of bundling actin [19]. The tau/f-actin interaction is also an important player in dendritic morphology and post-synaptic structure, therefore directly affecting synaptic stability [20]. Tau PTMs such as acetylation can modulate actin polymerization and cause synaptic dysfunction [21]. Formation of f-actin bundles is mainly produced by high-affinity interaction between proline-rich and microtubule-binding domains on tau protein and the filamentous actin. This is mediated by hydrophobic interactions between the tau microtubule-binding regions and actin filaments. On the other hand, an electrostatic bond also exists between tau proline-rich domain and f-actin. These offer a highly dynamic and multivalent interaction between tau and f-actin complexes, as the tau residues serve as “flexible linkers”. Structurally, one tau molecule contains seven actin-binding regions and these “flexible linkers” are located between these actin-binding sites [22]. Conformational changes in the actin-binding segments have a regulatory role for cross-linking between the three components of the cytoskeleton: tau, actin filaments and microtubules. Many aspects of the tau/f-actin complex, such as the exact binding sites, the bundling mechanism and the cross-linking between actin, tau and microtubules, are yet to be understood [22].

Neurons are structurally, morphologically and functionally distinct from any other cells. They have developed long processes and maintain an intricate functional connection through synapses with other proximal and distal target cells. This has evolutionarily forced them to have a sophisticated trafficking system for the transport of nutrients, energy and synaptic vesicles. “Fast axonal transport” is a complex intracellular trafficking process by which neurons transport the synthesized proteins in the somatodendritic compartment to the synapses or haul away the recycled materials and organelles. Axonal microtubules, through their molecular motors, kinesins and dyneins, play a crucial role in this bidirectional transport. While kinesin is used for transporting the cargo towards the cell periphery, dyneins are responsible for moving the cargo centrally towards the cell body, allowing for bidirectional traffic, navigating the crowded cytoplasmic environment and correctly conducting the transport tasks [23]. The connection between tau protein and microtubules can directly regulate this special ability of kinesin and dynein motors. This has been recently proposed to be mediated by the differential interaction of tau with different isoforms of dynein and kinesins. While kinesin-1 is more sensitive to inhibition by tau, kinesisn-2 and dynein are only inhibited at very high concentrations of tau. The overall sum of the connection between tau protein and microtubules regulates the forward movement or processivity of kinesin and dynein and therefore can affect the directional bias of axonal trafficking [24]. Of note, this kinesin-driven transport is tau phosphorylation-dependent, and inhibition of glycogen synthase kinase-3β (GSK3β) and reduction of tau phosphorylation is detrimental for axonal transport [25]. Increased GSK3β activity has been documented in AD, which can explain the hyperphosphorylation of tau and its detrimental effect on intracellular traffic in neurodegenerative diseases. In this active transport mode that is mediated by a tau-bound microtubule/actin network, tau is assumed to be stationary, although it is in a dynamic equilibrium with free tau molecules in the cytoplasm that are believed to be able to freely diffuse in the cytosol and axoplasm. This form of diffusion is one-directional and dependent upon the existing microtubule network. There is no need for external energy sources, and for short distances up to 1 μm, this form of tau diffusion is faster than motor-driven active transport. In this model, diffusing tau molecules occurs using the microtubule lattice to ensure proper distribution of tau molecules at different sites, i.e., somatic or axonal ends. This ensures that tau is always available for anterograde or retrograde transport systems [26].

Tau protein plays a critical role in post-synaptic scaffolding in dendrites. It interacts with the tyrosine kinase Fyn through its phosphatase activating domain (PAD), which is located on tau’s extreme N-terminal domain. The tyrosine kinase Fyn is responsible for phosphorylation of N-methyl-D-aspartate (NMDA) receptors, implying the impact of tau protein in synaptic signaling [27]. Moreover, in oligodendrocytes, tau-Fyn interaction affects the process outgrowth and is important in the initiation of axon myelination by oligodendrocytes [28].

Tau protein is not restricted to dendrites and axons, as it is also found in neuronal nuclei. In this compartment, tau serves as a DNA protection element against peroxidation through co-localizing with AT-rich heterochromatin regions of DNA and nucleoli; therefore, it contributes to the genomic stability and preservation of genomic organization [29,30,31].

## 4. Pathologic Tau

In 1906, Dr. Alois Alzheimer, a German psychiatrist and neuropathologist, described a five-year study on a clinical case with peculiar neuroanatomic features. His 50-year-old female patient suffered from paranoia, memory disturbances, sleep disorders and progressive confusion. After her death, Dr. Alzheimer investigated her brain autopsy and discovered intracellular neurofibrillary tangles (NFTs) and, consequently, described AD as an “unusual illness of the cerebral cortex” [32]. In 1963, these NFTs were characterized in the cortical neurons of the cerebrum in AD cases. These studies showed that NFTs are predominantly composed of insoluble fibers called paired helical filaments (PHFs) [33]. Two decades later, immunological assessments showed that “hyperphosphorylated tau aggregates” are the major components of PHFs [34]. Recent discoveries related to the tau protein and its pathological forms confirmed its key role in modulating neuronal physiology. To gain a better understanding of tau pathology, various aspects of this protein should be considered, including tau structure and distribution, its exact subcellular locations, possible posttranslational modifications and disease-specific isoforms.

## 5. One Protein and Various Conformers

Examination of tau aggregates obtained from different types of tauopathies revealed that tau filaments and inclusions are widely different in various types of tauopathies. These observations suggest that variations in tauopathy-related symptoms and also disease progression may be related to the specific pattern of tau aggregation. Tau filaments show astonishing variation in aggregation patterns, both in in vitro cell-free conditions and different tauopathy cases [35,36]. The predominant isoform of tau filaments in tau inclusions is used for the classification of different tauopathies (Figure 2) [37,38]. The complexity of different disease-specific tau structures may be a contributing factor in the complexity of tauopathies and finding treatments.

## 6. Abnormal Post-Translational Modifications of Tau

Tau protein is the target of many physiologic and pathologic post-translational modifications (PTMs). The real complex pattern of tau PTMs is not fully elucidated due to the fact that intermediate alterations in soluble human tau during the process of PHF formation and NFT aggregation cannot be visualized. Pathophysiologically, NFT formation requires the release of tau in a soluble (monomeric) form with a lower affinity for the microtubules than the physiologic tau. Hyperphosphorylation of specific tau residues strongly induces tau-microtubule dissociation, which leads to conformational changes. These alterations promote stable structures of anti-parallel tau dimers, oligomers and protomers that ultimately turn into PHFs and NFTs [39]. Structurally, there are 85 phosphorylation sites on tau, which are mainly serin, threonine and tyrosine residues [11]. Hyperphosphorylation of tau residues (244–368) located in the microtubule-binding domain can affect microtubule stability. In adjacent regions, phosphorylation of serin residues (Ser262 and Ser356) correlates with tau-microtubule dissociation [40]. In contrast, hyperphosphorylation of other residues, Ser214 and Thr231, can adversely affect tau affinity for the microtubules. Collectively, in full-length tau, there are 45 serins, 35 threonine and 5 tyrosine residues that are prone to hyperphosphorylation [41] (Figure 3). This adds another layer of complexity to the heterogeneity of tauopathies.

Hyperphosphorylation of tau in neurons reflects abnormal activity of protein phosphatases and/or dysregulation of kinases. According to the published literature, alterations in the levels of active kinases are associated with tau hyperphosphorylation in tauopathies. For instance, aberrant activation of cyclin-dependent-like kinase 5 (CDK5), glycogen synthase kinase-3β (GSK3β) and its regulator, c-JUN N-terminal kinase (JNK), are associated with NFT formation [42,43,44]. The dysfunction of protein phosphatase 2A (PP2A) has also been identified as another activator of tau hyperphosphorylation. Reduced PP2A expression levels and upregulation of PP2A inhibitors closely correlate with PP2A deregulation mechanisms in tauopathies. PP2A directly enhances tau phosphorylation by preventing tau dephosphorylation and indirectly by upregulating tau kinases [45]. 

In addition to phosphorylation, tau can undergo other PTMs, including glycosylation, glycation, nitration, truncation, ubiquitination, acetylation and methylation. The pie chart below, generated from the available literature, represents the proportion of PTM-related residues on tau protein (Figure 4) [46,47]. Additionally, abnormal acetylation and ubiquitination of tau can also contribute to neurodegeneration [48].

Oxidation is another form of tau PTMs that has been identified as the inducer of tau fibrilization under oxidative stress conditions. This can affect five tyrosine residues in the 2N4R tau isoform, at positions 18, 29, 197, 310 and 394 [49]. Under oxidative stress, these tyrosine residues are prone to the formation of dityrosine (DiY) cross-links, which prevent tau elongation and form trapped, stable and insoluble tau species, by which NFT formation can be promoted [50].

Post-translational modification of tau, particularly tau phosphorylation and acetylation, can trigger a tau condensation mechanism known as liquid-liquid phase separation (LLPS) in neurons. This mechanism is reported in both physiological and pathological conditions and is regulated by electrostatic interactions between oppositely charged regions of tau isoforms or between tau proteins and RNA. LLPS leads to the formation of liquid droplets containing high levels of concentrated tau, which can electrostatically interact with other tau isoforms or RNA molecules. Although the mechanistic details of LLPS are poorly understood in neurons, it has been shown to accelerate neuronal tau fibrilization and facilitate neurotoxicity in tauopathies [51,52]. 

## 7. Inducers of Pathologic Tau, Upstream Mechanisms

There are approximately 57 neurodegenerative-related mutations on the MAPT gene, which were mainly discovered in pedigrees with familial frontotemporal lobar degeneration (FTLD). Predominantly, inherited mutations in the MAPT gene (FTDP-17) are associated with neuronal and glial tau inclusions in patients with FTLD [53]. The specific mutation site affects the outcome, as shown for mutations in exon 10 that can affect the microtubule-binding affinity of the protein and change the ratio of 3R:4R in adults. Even intronic mutations in MAPT (e.g., intron 10) may affect tau protein structure and the isoform ratio, resulting in pathogenic tau formation [54,55]. To date, no association between MAPT mutations and PTMs in tau protein has been discovered [56].

The second identified cause of pathogenic tau is chronic traumatic encephalopathy, which is also listed as a type of progressive neurodegenerative tauopathy. Mechanistically, brain injury can induce tau cleavage, its aberrant phosphorylation and aggregate formation. These processes can adversely affect axonal microtubule organization, resulting in tau-microtubule dissociation and subsequent tau accumulation in axons. The key link between brain injury and tau hyperphosphorylation is attributed to the reduction in alkaline phosphatase levels, which is mainly responsible for tau dephosphorylation. All these pathophysiological conditions can ultimately trigger chronic inflammation, apoptosis and neuronal degeneration in brain tissue [23]. Accumulation of pathologic tau also occurs as part of normal aging, as it has been detected in the temporal neocortex of people over the age of 65. This condition, known as “primary age-related tauopathy”, is a normal senescence feature in primates [57].

In addition to the aforementioned stimulators, pathogenic tau can also be induced in people with metabolic syndrome. Examination of in vitro and in vivo reports as well as results from post-mortem human patients depict a strong relationship between tauopathy and neurons containing increased levels of phosphorylated insulin receptor substrate-1. This results in impaired insulin signaling, which can drive pathogenic tau aggregation through reducing phosphoinositide 3-kinase–Akt activity and activating GSK3β. This in turn induces tau hyperphosphorylation and its dissociation from microtubules [58] (Figure 5).

Additionally, pathogenic tau formation is shown to be triggered through neuron-to-neuron transmission in a prion-like manner. Various mechanisms are identified as the stimulators of this type of pathogenic tau aggregation. For instance, the pathogenic tau spreads trans-synaptically from one cell to another or through “exosomes” which are present in CSF and plasma [59,60]. These secretory extracellular microvesicles that contain hyperphosphorylated tau aggregates might be released from the affected neurons or microglia [61]. The other factor in this type of pathogenic tau propagation are ectosomes: larger vesicles than microvesicles, which are packaged in the affected cells via plasma membrane budding. These vesicles participate in the pathological spreading of tau protein [62].

The mechanistic target of rapamycin (mTOR) has been identified as another potential driver of pathogenic tau formation. This serin/threonine protein kinase plays a crucial role in neuronal growth and maintenance. In neurons, mTOR regulates prominent cellular functions including autophagy, signal transduction, metabolism and cytoskeletal dynamics [63]. Hyperactivation of mTOR can induce tau expression and abnormal tau phosphorylation. It has also been revealed that mTOR activation can adversely affect autophagy rate, which subsequently interferes with pathogenic tau clearance in neurons [64].

## 8. Pathologic Tau and Cell Toxicity, Downstream Events of Tau Dysfunction

The association of pathogenic tau with neuronal toxicity and disruption of neuronal networks, as well as glial damage, has been well documented in the literature. However, the exact mechanisms and physiological events that link pathogenic tau formation to neurotoxicity and cell death remain unknown. As AD is well-characterized microscopically by the combined presence of intracellular NFT aggregates and extracellular Aβ plaques, inevitably, the tauopathy-related studies are vastly focused on AD models. The impact of tau pathology on the cellular cytoskeleton and its negative effect on intracellular traffic of synapse machinery were discussed previously. We know that hyperphosphorylated tau can stabilize dendritic microtubules and induce synaptic dysfunction [65]. These peri-synaptic alterations induce synaptic malfunction by adversely affecting synaptic anchoring and glutamate receptor trafficking [66]. In addition to the peri-synaptic alterations, pathogenic tau might disrupt dendritic spines. Recent proteomic data obtained from an in vivo study on Tau-P301S transgenic mice demonstrated that several proteins and pathways may change in post-synaptic densities (PSDs). For instance, the levels of GTPase-regulatory proteins, which participate in cytoskeletal actin dynamics and dendritic spine stability, may decline significantly, although synapse loss is identified as an early event and direct consequence of tau pathology. How exactly pathogenic tau induces synapse loss is yet to be understood [67].

Impaired axonal transport is another identified pathologic event that is attributed to tau toxicity. “Long-range transport” in neurons is mediated by kinesin motors, which can be inhibited by tau protein. Mechanistically, tau can limit cargo travel distance by reducing kinesin velocity [68]. Under pathologic conditions in the soma, tau traps the kinesin adaptor molecule c-JNK-interacting protein 1(JIP-1) and prevents long-range transport. On the other hand, kinesin deficiency can trigger tau hyperphosphorylation, which further exacerbates the transport efficiency [69].

The impact of oxidative stress in neurodegenerative diseases, including AD, has been well recognized; however, it is not clear whether tau-aggregation results in excessive production of reactive oxygen species (ROS) or whether tau hyperphosphorylation is mediated by ROS levels. It has been shown that neurons containing tau aggregates produce high levels of ROS and externalize a considerable amount of phosphatidylserine. Microglial cells can detect the aberrant levels of phosphatidylserine and phagocytose both the affected neurons and tau aggregates. Additionally, microglia can release opsonin milk-fat-globule EGF-factor-8 (MFGE8) and nitric oxide (NO), which helps facilitate the engulfment process [70]. In another study, the leading role of ROS was suggested by the induction of tauopathy in the *Drosophila* model, in which downregulation of thioredoxin-1 or superoxide dismutase was associated with increased tau hyperphosphorylation [71]. Activation of JNKs has been reported after decreased thioredoxin activity [72,73].

Tau-mediated neurotoxicity can also be mediated by the overstabilization of actin filaments, which ultimately disrupts the cytoskeleton and synaptic plasticity. Additionally, recent findings demonstrate that pathologic tau formation triggers f-actin stabilization, weakens nuclear pore localization/function and disrupts nucleocytoplasmic trafficking [74,75]. Consequently, f-actin bundling can also affect mitochondrial dynamics and induce elevated levels of ROS [76]. Some evidence suggests the involvement of epigenetic changes with tau hyperphosphorylation. Overstabilization of f-actin has been shown to exert pressure on the nuclear envelope, resulting in invagination of the nuclear envelope and proteolytic degradation of the nuclear lamina. Structurally, the nuclear lamina serves as a strong scaffold for the DNA and is important for the DNA heterochromatin/euchromatin ratio. Damage to the nuclear lamina will result in changes in heterochromatin relaxation and transcription of the normally dormant/silenced regions of DNA. An elegant example of these changes is the re-expression of cell cycle entry-related proteins in postmitotic and fully-differentiated neurons, which is a strong signal for apoptotic cell death [77]. In parallel with chromatin changes, the expression patterns of RNA are also affected, which may promote RNA instability and activate transposable elements [77]. Overexpression of tau in *Drosophila* and mouse models of tauopathy was shown to activate the normally silenced retrotransposons. These results were also confirmed in the post-mortem brains obtained from AD and progressive supranuclear palsy (PSP) patients [78,79,80] (Figure 5).

## 9. Diagnosis and Therapeutic Strategies for Tauopathies

Accurate diagnosis of tauopathies has been made possible by tau positron emission tomography (PET) scans. A tau-PET scan allows for the detection of tau deposition in the patient’s brain. Using this technique allows for the detection of tauopathy severity and progression and can also be used for monitoring the efficacy of therapeutic interventions. A growing list of tau-PET tracers is available for clinical examination of tauopathy, among which THK5317, THK5351 and PBB3s are used for clinical assessments [81]. Additionally, [^18^F] GTP1, [^18^F] flortaucipir, [^18^F]MK-6240, and [^18^F]RO948 tau-PET have been studied as other tau tracers [82]. [^18^F]-flortaucipir (AV1451) is a first-generation FDA-approved tau tracer that strongly correlates with identifying NFT distribution in AD [1]. 

The complexity and multifactorial nature of tauopathies is the major obstacle to better understanding the pathology and designing new therapeutic interventions for slowing the disease’s progress. The current clinical tools cannot be used for the detection of early-stage diseases, and the diagnosis can only occur in the late stages, when the currently prescribed drugs can only treat the symptoms. There is an urgent need for developing early diagnosis tools and developing disease-modifying drugs and, accordingly, many preclinical and clinical trials are currently in progress. Since AD is the most common form of tauopathy, the efforts to discover novel therapeutic interventions have also been mostly focused on this disease. However, considering the complexity of tau isoforms and inclusion bodies, it is not clear whether an effective drug for AD-associated tauopathies can also cure other forms of tauopathies. Whether treatments targeted for tauopathies may need to be personalized remains to be investigated. The timing of the application of any potential treatments represents the next challenge in human diseases. Notably, the pathologic events in tauopathies, including tau hyperphosphorylation and protein aggregation, occur a long time before the possibility of a clinical diagnosis, when experimental treatments are proven to be ineffective. These limitations urge the need for the discovery of novel biomarkers to allow diagnosis at earlier stages of the diseases. Two categories of tau biomarkers have been identified in CSF and blood. In CSF, increased levels of phosphorylated tau at residue 217 (p-tau217) and p-tau235 as well as p-tau231 are shown to be correlated with the early stages of AD [83], while p-tau181 can be used as a prognostic tool for the advanced phases of AD [84]. The presence of the axonal protein neurofilament light chain (NfL) in CSF has also been identified as a promising diagnostic tool for AD and FTLD [85]. Searching for suitable CSF biomarkers has various limitations, as obtaining CSF is an invasive approach that can be performed only in more advanced clinical centers, and therefore it is not very cost-effective to be used for all patients [86]. Identification of plasma biomarkers such as p-tau181 [87] and p-tau217 [88] in peripheral blood can be an alternative and reliable biomarker for AD. Interestingly, plasma p-tau217 is specific for AD and is not detected in non-AD tauopathies, which makes it a useful biomarker to discriminate AD from the other forms of tauopathies [89]. Critiques of plasma biomarkers argue that they may not properly reflect the CNS pathophysiological changes since these proteins are degraded in the kidney and liver, which may affect their reliability [86].

Numerous studies have demonstrated evidence of considerable achievement in designing therapeutic interventions for tauopathies. Proper identification of druggable pathophysiological mechanisms in tauopathies is fundamental to this achievement. Some of the potential mechanisms that can be targeted for this purpose include: (1) tau PTMs, particularly tau hyperphosphorylation, by phosphatase modifiers and kinase inhibitors, (2) targeting cytoskeleton/nuclear disruption and genomic architecture dysfunction and (3) reducing tau expression levels with small interfering RNA (siRNA) and antisense oligonucleotides (ASOs). Additional approaches include reducing the levels of plasma tau, inhibiting the aggregation of different forms of pathogenic tau such as PHFs, NFTs and small oligomeric tau aggregates, as well as enhancing tau clearance by active and passive tau immunization [90] (Figure 6).

Passive immunotherapy has been the focus of tau clinical trials, with the aim of slowing the neurodegenerative process through preventing the spread of pathogenic tau and removing its extracellular aggregations. The main barrier to finding an effective treatment for clinical purposes is finding a suitable epitope on tau protein that is commonly present in all patients despite their genetic background or the specific type of tauopathy disease. Active tau immunotherapy has also been explored. AADvac1 is a therapeutic tau vaccine candidate that has shown successful reduction of neurofilament light chain (NfL) in plasma and CSF in AD patients by recognizing a 12-amino acid sequence (KDNIKHVPGGGS) in tau MTBR. Reported results from its phase I clinical trial have raised hope for tauopathy treatment. According to these results, AADvac1 leads to reduced cognitive decline and lower brain atrophy in AD patients in mild to moderate stages [91,92,93]. Obtained results from its phase II trials confirmed its safety and well-tolerance; however, it was not completely successful in cognition improvement [94]. The other candidate tau-targeted vaccine for patients with early AD is ACI-35. This immunotherapy vaccine can target the pathological tau phosphorylation at S396 and S404. This phase I/II clinical trial is expected to be completed before October 2023 [94,95]. Among the wide range of tau therapeutic approaches, clinical trials for developing effective tau antisense oligonucleotides (ASO) therapeutics are currently in progress. ASOs are short, single-stranded DNA-like molecules synthesized to target tau mRNA in a highly selective mode and, therefore, can reduce both intercellular and extracellular tau levels. While ASOs are highly selective and safe, their limited permeability across the blood–brain barrier (BBB) is a major limiting factor, although this can be overcome by intrathecal administration [96]. IONIS-MAPTRx (BIIB080) is the first clinical trial of tau ASOs, which aim to reduce MAPT mRNA in patients with mild AD [97]. A list of the most recent therapeutic interventions for tauopathies is presented in Table 1.

## 10. Summary and Conclusions

Dementia is an umbrella term for many neurodegenerative diseases. As we age longer, the rate of dementia is exponentially increasing, imposing a significant emotional cost on the patients’ families and a financial burden on human societies. Tauopathies are a major contributor to dementia, and despite extensive research, the exact underlying neurodegenerative mechanisms remain to be elucidated. Tauopathy diseases present extensive neuropathological and phenotypical differences, which impose considerable challenges in developing diagnostic and therapeutic strategies. Despite significant transformative achievements in the field of tauopathy, several fundamental questions remain to be addressed. Novel drugs are in clinical trial phases with some promising outcomes; however, proving their safety and efficacy will require large human studies that may take several years to complete. Preclinical studies in *Drosophila* and mouse models of tauopathy have revealed a new aspect of tauopathy that is characterized by nuclear envelope invagination and changes in gene expression. Interestingly, nuclear envelope invagination has also been confirmed in human post-mortem brain tissue, which results in the re-emergence of endogenous ancient retroviruses. Calorie restriction in *Drosophila* effectively attenuated the endogenous retroviral expression [71]. The protective effect of calorie restriction has been related to decreases in tau hyperphosphorylation in *Drosophila* [71] and in the ApoE-deficient mouse model of neurodegeneration [117] as well as in the Tg4510 mouse model of tau deposition [118]. While there is no information on similar impacts in humans, these data indicate that changes in lifestyle may be an effective preventive way to decrease the impact of age-associated diseases. With the application of new imaging tools such as Tau-PET tracers, it will be possible to diagnose patients in earlier stages of the disease, which may respond better to the available treatments. Alongside these changes, the need for the identification of new biomarkers is a high priority that will ultimately improve patients’ responses to drugs and their quality of life.

## Figures and Tables

**Figure 1 biology-12-00244-f001:**
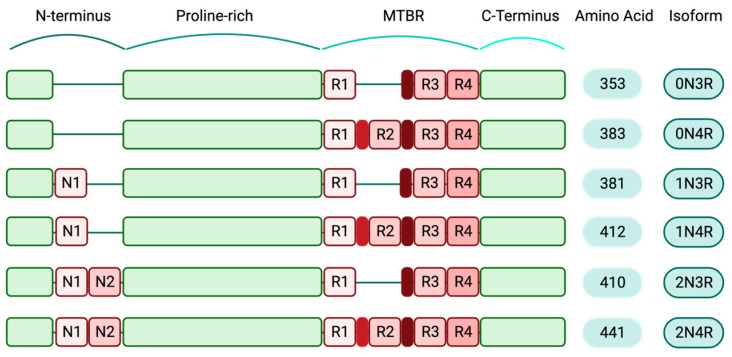
Human-specific tau isoforms (created with BioRender.com). Schematic diagram depicting the composition of different tau isoforms. The shortest tau variant, which is mainly expressed during the human embryonic stage, consists of 352 amino acids (0N3R) and the full-length isoform (2N4R) includes all structural domains of tau protein and 441 amino acids. Structurally, tau isoforms are comprised of N-terminus regions that contain two inserts, N1 and N2. The au N-terminals are followed by the proline-rich domains, the microtubule-binding regions (MTBR) and the C-terminals. In the full-length tau isoform, all four repeat domains (R1–4) are present in the MTBRs.

**Figure 2 biology-12-00244-f002:**
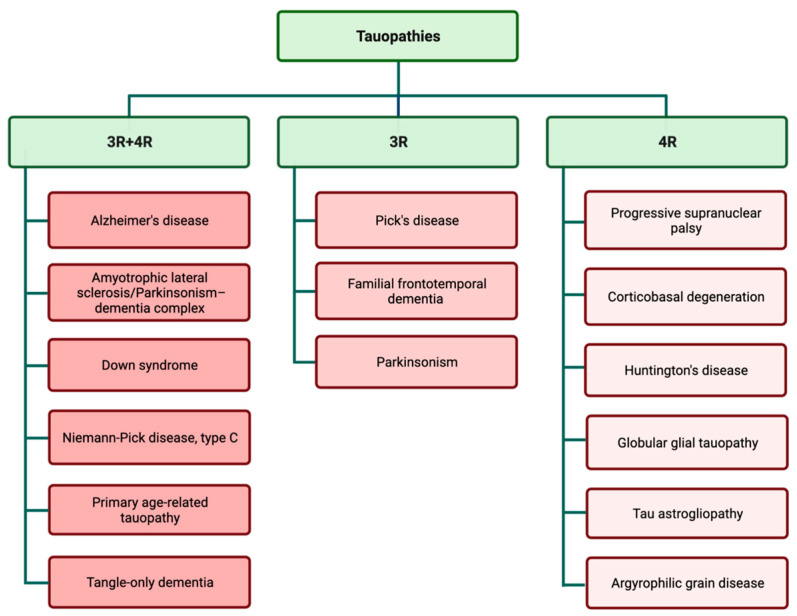
Tauopathies are classified according to the presence of 3R and 4R isoforms in tau inclusions.

**Figure 3 biology-12-00244-f003:**
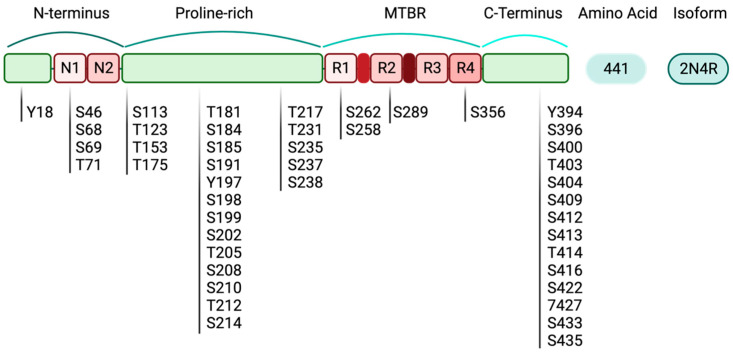
Tau phosphorylation sites that are predominantly located in the proline-rich region and C-terminal domain of the 2N4R isoform. In full-length tau, there are 45 serins (S), 35 threonine (T) and 5 tyrosine (Y) residues that are prone to hyperphosphorylation.

**Figure 4 biology-12-00244-f004:**
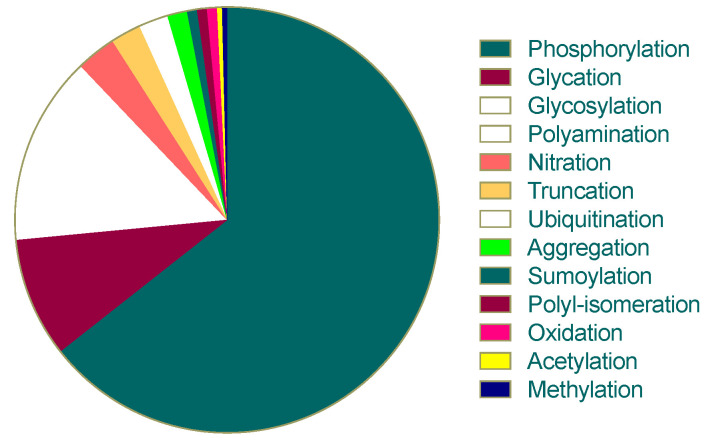
Proportion of PTMs on tau protein according to the number of targeted residues in the tau 2N4R isoform.

**Figure 5 biology-12-00244-f005:**
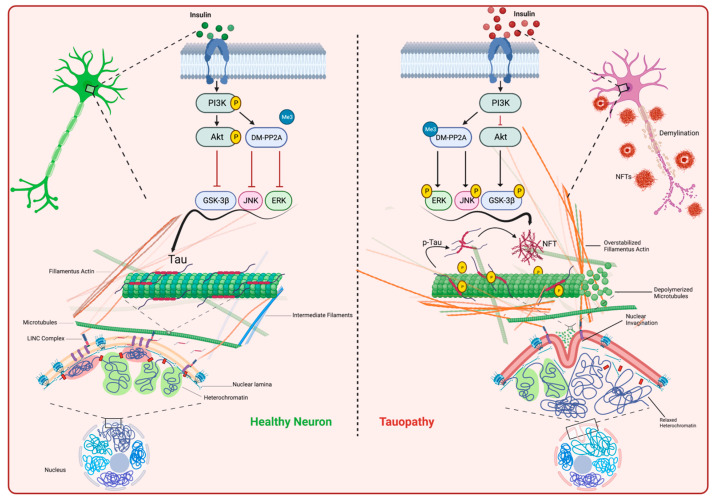
Pathogenic tau, upstream and downstream events (created with BioRender.com). In healthy neurons (**left**), physiologic tau is responsible for microtubule assembly and maintenance of critical cellular components such as the cytoskeleton, the nucleoskeleton and DNA integrity. Under pathologic conditions (**right**), PTMs can alter tau functions. For instance, activated ERK, JNK or GSK3β induces tau hyperphosphorylation, which is the main trigger of tau aggregation and NFT formation in the neurons. Consequently, this may lead to microtubule disassembly, actin overstabilization and nuclear lamina invagination. These circumstances can dysregulate genomic DNA integrity. Moreover, extracellular release of NFTs facilitates their neuron-to-neuron transmission. These pathophysiological events ultimately result in progressive neurodegeneration.

**Figure 6 biology-12-00244-f006:**
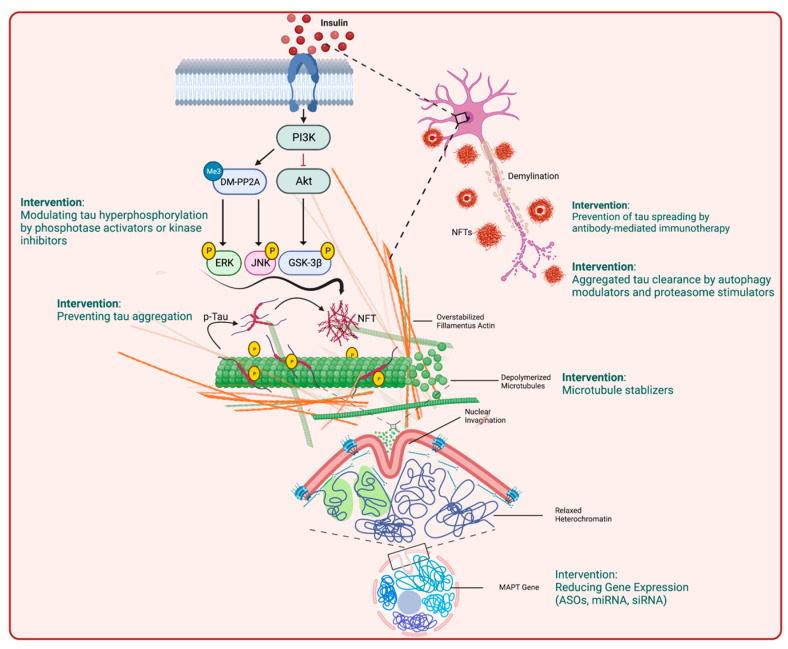
Tau-related therapeutic interventions (created with BioRender.com). Clinical trials and preclinical studies on tauopathies mainly focus on various aspects of pathogenic tau and its consequences, including modulating tau hyperphosphorylation and inhibiting other PTMs such as O-deglycosylation, preventing tau aggregation or stimulating tau clearance by autophagy modulators or proteasomal degradation, active or passive immunotherapies, applying microtubule stabilizers and reducing MAPT gene expression levels.

**Table 1 biology-12-00244-t001:** Recent preclinical and clinical therapeutic interventions for tauopathies.

Model/Cell Line	Therapeutic Intervention	Intended Mechanism	Potential Target	Reference
Human patients with probable AD or MCI-AD	Hydromethylthionine mesylate	Inhibiting tau aggregation by targeting pathological tau oligomers and filaments In the Phase III clinical trial.	Tau aggregation	[98]
Human patients with mild to advanced AD	Cerebrolysin^®^ & Donepezil	Unknown	Tau expression and tau phosphorylation	[99]
Prodromal to mild AD patients	Semorinemab	Not slowing tau accumulation pathology (not effective), and no change in clinical AD progression.	Oligomeric tau	[100]
Cyno monkeys, C57Bl/6J mice model, PAC transgenic mice, hESC line SA001 cell line	ASO-001933	Selective and long-lasting reduction in tau levels by locked nucleic acid (LNA)- modified ASOs.	Tau aggregation	[101]
3 × Tg mice model	Norvaline	Diminishing tau phosphorylation levels.	Phosphorylated tau	[102]
Murine polytrauma mouse model	Propranolol	Decreasing hippocampal p-tau accumulation.	Accumulation of phosphorylated tau	[103]
Ty1-hTau.P301S mice and SH-SY5Y cell line	Tetrandrine	Promoting tau clearance and degradation via autophagy, rescuing lysosomal Ca^2+^ homeostasis, and diminishing NFT development.	Lysosomal two-pore channel 2 (TPC2)	[104]
Htau and JNPL3 mouse models	Tau oligomer monoclonal antibodies (TOMAs)	Reducing tau oligomer levels by tau passive immunotherapy.	Tau oligomeric strains	[105]
P301S mouse model	Glimepiride	Decreasing GSK3β, increasing phosphorylated-AKT/total-AKT, increasing PP2A, and normalizing CDK5 levels. Decreasing neuroinflammation and apoptosis by reducing NF-kB, TNF-α and caspase 3 levels	Phosphorylated tau	[106]
Human AD patients	Donanemab	Slowing tau accumulation	Tau accumulation	[107,108]
PS19 mouse model	Etanercept and TfRMAb-TNFR	Reducing phosphorylated tau and microgliosis, increasing PSD95 expression and attenuating hippocampal neuron loss.	TNF-α (Inhibitors)	[109]
AAVhTau mouse model	Dihydroartemisinin (DHA)	Inducing tau O-Glc-N-Acylation modification, reducing tau phosphorylation, improving learning and memory and increasing hippocampal CA1 long-term potentiation (LTP).	PTMs on tau protein	[110]
Tau-P301S mouse model	Rutin	Inhibiting tau aggregation and its oligomer-induced cytotoxicity, reducing the production of proinflammatory cytokines and preserving neurons.	Tau aggregation	[111]
rTg4510 mouse model	BSc3094	Reducing tau phosphorylation, improving cognition and reducing anxiety-like behavior.	Tau aggregation	[112]
Neuroblastoma cell model (with methyl glyoxal (MG)-induced Tau glycation)	Epigallocatechin-3-gallate (EGCG)	Inhibiting glycation, modulating tau phosphorylation, enhancing actin-rich neuritic extensions and preserving the actin and tubulin cytoskeleton.	Cytoskeleton stabilizer	[113]
SH-SY5Y and HEK293 cell lines hTau-transgenic, tauP301L and 3 × Tg-AD mouse models	C004019	Promoting tau ubiquitination-proteasome-dependent proteolysis Improving synaptic and cognitive functions in animal models.	Tau clearance	[114]
3 × Tg-AD mouse model	Intravenous administration of mesenchymal stem cells	Decreasing pathological tau phosphorylation at T205, S214, T231 and S396 but not levels of Aβ-42.	Phosphorylated tau	[115]
Human PSP patients	Gosuranemab	Decreasing unbound N-terminal tau in CSF. In spite of this effect, gosuranemab did not show clinical efficacy in PSP patients. Phase II clinical trial is completed.	N-terminal tau	[116]

## Data Availability

Not applicable.

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
