# Peer review of "Tau; One Protein, So Many Diseases"

_biology, 2023, doi:10.3390/biology12020244_

Round 1

Reviewer 1 Report

The authors provide a comprehensive overview of the physiological and pathological roles of tau protein in various neuronal functions and associated diseases. Extensive data from the literature are reported and discussed in well-organized sections of the manuscript and summarized graphically in relevant figures and tables. The manuscript is a review article on tau protein and, as such, does not answer specific questions, but aims to provide a compendium of knowledge on the functional and pathological role of tau protein. Specifically, the manuscript addresses the  mechanisms that lead to the  formation of the pathologic form and the mechanims by which the pathological tau causes neurotoxicity. Therapeutic approaches and diagnostic methods used so far are reviewed and new methods are proposed. 

The topic is relevant to the biomedical field, as it highlights possible therapeutic strategies based on knowledge gained to date.

The manuscript has the merit  of organizing  a large amount of literature data into different sections with the aim of highlighting both general and specific aspects related to the relationship between different isoforms of tau protein and diseases, the pathologic role of hyperphosphorilation and other Post-translational Modification with reference to the specific residues involved, and the main therapeuthic approaches used in the treatment of different diseases. Both figures and tables are helpful in presenting this overview.
  The summary and conclusions are consistent with the purpose of the manuscript   Bibliographic references are relevant and appropriate, including the most relevant previous and recent literature   

Author Response

Dear Reviewer,

Thank you very much for your valuable time, consideration and your positive comments on our work.

Regards,

Parisa Tabeshmehr

Reviewer 2 Report

Tau protein is a microtubule-associated protein in neurons and in many neurodegenerative diseases (collectively known as tauopathies) tau self-assembles to form distinct tau filaments. In this review paper, the authors provided a brief but helpful overview on tau protein, describing the most important biological functions of tau and highlighting the pathological role of tau in tauopathies. They also provided a list of some recent preclinical and clinical trials for tauopathies. Overall, this is a well written and structured manuscript, I do however have some comments for the authors that I believe will improve the manuscript. 

1.     Many statements in the manuscript were not supported by references, authors must add references to the statements ending in lines number 85, 118, 138, 154, and 272

2.     Liquid-liquid phase separation as a mechanism of pathological tau in tauopathies should be added and discussed.

3.     “Tyrosine” in missing in line 176 “kinase Fyn”

4.     Many studies showed that tau protein can undergo oxidative modification to form dityrosine crosslinked oligomers and recently dityrosine crosslinks were detected in tau NFTs in human AD brain (Maina, MB., et al.,  Dityrosine Cross-links are Present in Alzheimer’s Disease-derived Tau Oligomers and Paired Helical Filaments (PHF) which Promotes the Stability of the PHF-core Tau (297–391) In Vitro, Journal of Molecular Biology, 2022), I think this should be added/discussed in abnormal PTMs of tau section. 

5.     In Figure 4, same colours were used in the pie chart to represent the proportion of some PTMs such as glycosylation, nitration, truncation, ubiquitination, aggregation and oxidation. Could you please use different colours? 

Author Response

Dear Reviewer,

Thank you very much for your valuable time and constructive comments on our work. Here, I listed the point-by-point response to your comments.

Comment 1: I added all those references and highlighted the reference numbers.

Comment 2: LLPS in pathology of tau is added in tau PTMs section and is highlighted in blue.

Comment 3: the word "Tyrosine" is added to the requested line and highlighted in blue.

Comment 4: I added a paragraph related to Dityrosine Cross-links in tauopathies and you can find it in Tau PTMs' section, highlighted in blue.

Comments 5: I tried to change the colors in the pie chart but I'm not sure if they are close to what you wanted. Hopefully, they are.

Finally, thank you again for all your comments and "please see the attachment"

Regards,

Parisa Tabeshmehr
